# Effects of Laser Hardening Process Parameters on Hardness Profile of 4340 Steel Spline—An Experimental Approach

**Noureddine Barka [1,\*], Sasan Sattarpanah Karganroudi [2], Rachid Fakir [1], Patrick Thibeault [1] and Vincent Blériot Feujofack Kemda [1]**

[1] Mathematics, Computer Science and Engineering Department, Université du Québec à Rimouski, Rimouski, QC G5L 3A1, Canada; rachid.fakir@uqar.ca (R.F.); Patrick.thibeault@uqar.ca (P.T.); bleriotvincent.feujofackkemda@uqar.ca (V.B.F.K.)

[2] Institut Technologique de Maintenance Industrielle, Cégep de Sept-Iles, Sept-Iles, QC G4R 5B7, Canada; sasan.karganroudi@itmi.ca

[\*] Correspondence: noureddine_barka@uqar.ca

**Abstract:** This study displays the effect of laser surface hardening parameters on the hardness profile (case depth) of a splined shaft made of AISI 4340 steel. The approach is mainly based on experimental tests wherein the hardness profile of laser hardened splines is acquired using micro-hardness measurements. These results are then evaluated with statistical analysis (ANOVA) to determine the principal effect and the contributions of each parameter in the laser hardening process. Using empirical correlations, the case depth of splined shaft at tip and root of spline's teeth is also estimated and verified with measured data. The obtained results were then used to study the sensitivity of the measured case depths according to the evolution of laser process parameters and geometrical factors. The feasibility and efficiency of the proposed approach lead to a reliable statistical model in which the hardness profile of the spline is estimated with respect to its specific geometry.

**Keywords:** AISI 4340 steel; analysis of variances; case depth; laser hardening; splined shaft; experimental analyses

## 1. Introduction

Laser surface hardening is a heat treatment process that uses a high energy density laser in a localized heating zone [1,2]. Applying this process on steel parts and coupling it with a fast quenching technique, called martensitic quenching, allow improving fatigue life and limit the distortion of steel parts [3–5]. The laser beam is capable of rapidly increasing the surface temperature to above the austenitizing temperature ($Ac_3$). This intense heating process makes it possible to transform the existing microstructure of the surface to austenite, and then to generate a very fine martensitic microstructure with high hardness [6]. In order to obtain a uniform hardness profile on parts with complex geometries, specific tuning on the parameters of laser hardening processes, such as laser power and scanning speed, is required [7–9]. This uniform hardness profile is recommended to uniformly distribute the mechanical loads. This layer is obtained by a uniform martensitic layer at the surface of hardened parts. This uniform hardened layer avoids stress concentration on the hard surface layer of parts and limits crack propagation under periodic loads, which improves the fatigue resistance of parts [10–12]. In the case of splines, a uniform hardness profile on teeth geometry can effectively reduce the wearing caused by abrasion [13].

To achieve this uniform hardness profile, it is important to calibrate parameters such as laser power, scanning speed, and rotational speed. Sun et al. [14] proposed an experimental approach

supported with an analysis of variance (ANOVA) to develop a model for optimizing laser cladding parameters. This approach allows demonstrating an optimal combination of process parameters to obtain a high-quality microstructure with laser coating applied to Ti6Al4V. Lambiase et al. [15] proposed a prediction method that can determine the temperature distribution profile based on laser hardening process parameters using an artificial neural network technique. The artificial neural network model allows for a better understanding of laser heating process behavior and the validation of simulated data. Bailey et al. [16] carried out tests to optimize the laser hardening processes of industrial parts with complex geometries by developing a predictive model. Their approach can estimate the amount of heat generated during the process and transferred to the surface of a splined shaft. Thus, it can predict the hardness profile based on the heated area on the surface layer of parts. Babu et al. [17] carried out experimental studies on the microstructure of low alloy steels wherein the durability after laser hardening is studied. Their approach allows optimizing the hardness profile of EN25 steel and obtaining a surface hardness twice as high as the core hardness by varying the laser power and sweeping speed. Lusquinos et al. [18] carried out theoretical and experimental studies of laser heat treatment applied to 1045 steel using a high-power diode laser source. The results of their study demonstrate that obtaining a greater degree of hardness can be archived by maintaining a relatively constant surface temperature during the process.

Most of the previous research works were carried out on simple geometries. However, in this paper, we present a comprehensive set of experimental tests on a complex geometry of a splined shaft and present a statistical approach to evaluate the effect of the laser hardening process and geometrical parameters on the case depth of splines. To this end, the first step is to plan experimental tests by varying different parameters of laser hardening procedures and geometrical parameters of splines. The second step involves defining the variation range of each independent parameter. The third step consists of acquiring the experimental results and analyzing these data using ANOVA analysis [19,20]. The final step is dedicated to developing a reliable statistical model to estimate the case depth hardness of splines, according to their geometry.

This article is organized as follows. Section 2 presents the experimental planning, the methodology, and a description of the proposed approach toward using statistical analysis to determine the effect of laser hardening parameters on splined shafts. Results of experimental tests as well as statistical and sensitivity analyses are presented in Section 3. The paper ends with a conclusion and ideas for future works on this domain in Section 4.

## 2. Methodology and Experimentations

The experimental tests presented in this article are carried out on AISI 4340 steel splines. The chemical composition of AISI 4340 Steel and its mechanical characteristics are given in Tables 1 and 2, respectively. The geometrical parameters of these splines, as shown in Figure 1, consist of a diameter of 29 mm, flank tilt angles of 15° and 20°, and the tooth depth of splines, which varies in three different depths: 2.5, 3, and 3.5 mm.

**Table 1.** AISI 4340 steel chemical composition as a percentage.

| C | Cr | Fe | Mn | Mo | Ni | P | S | Si |
|---|----|----|----|----|----|----|----|----|
| 0.37–0.43 | 0.7–0.9 | 96 | 0.7 | 0.2–0.3 | 1.83 | <0.035 | <0.04 | 0.23 |

**Table 2.** Mechanical properties of AISI 4340 steel.

| Yield Strength [MPa] | Ultimate Tensile Strength [MPa] | Elongation at Break [%] | Hardness Rockwell C [HRC] |
|---|---|---|---|
| 710 | 1110 | 13.2 | 35 |

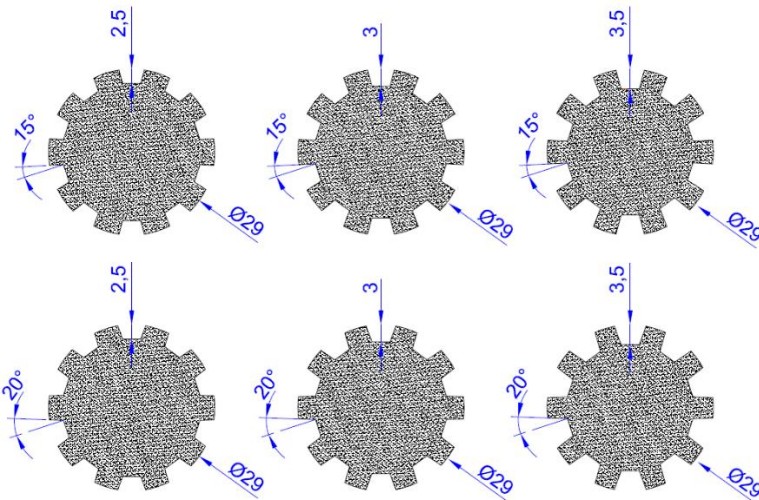

**Figure 1.** Geometric variation of experimented spline shafts, dimensions are in [mm], and angle is in degrees [°].

Before applying the laser hardening heat treatment, the spline shafts are quenched and tempered to achieve a uniform core hardness of approximately 45 HRC. The experimental laser hardening set-up, as illustrated in Figure 2, is composed of a 3 kW fiber laser source and six degrees of freedom (DF) articulated FANUC robot, which supports the laser head. A rotating rig is also integrated into the experimental set-up, which provides a maximal rotation speed of 10,000 rpm. This experimental set-up is also equipped with a mixed mounting system, comprising central support with three jaws and a counter-head that moves along the z-axis. This mounting system allows centering the spline shafts during rotation. The laser spot is located in the median plane of splines, which are laser heat-treated. Following laser heat treatment, the samples are carefully prepared, polished, and etched using a Nital chemical solution (95% ethanol and 5% nitric acid). The hardness profiles are then characterized by micro-hardness measurements using a Clemex machine.

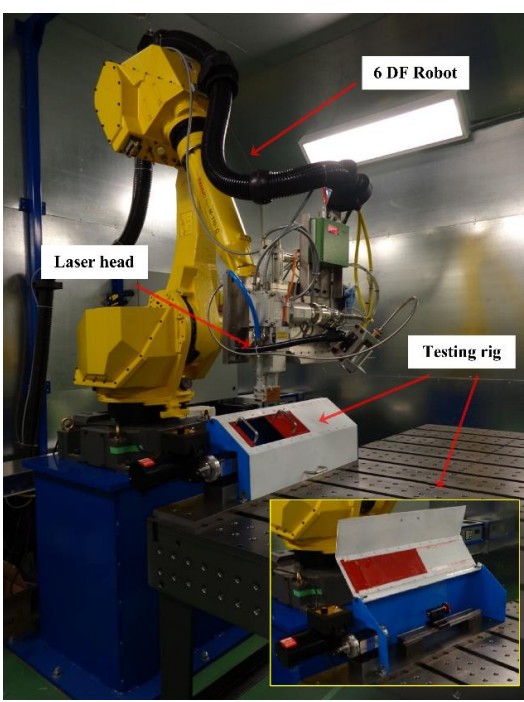

**Figure 2.** Experimental set-up equipped with a robotic laser cell [21].

To evaluate the case depth at the tip and root of spline teeth, a series of experimental tests are carried out wherein the laser hardening process and splines' geometrical parameters are varied. Adopting an experimental plan in which different parameters vary within their variation range, allow studying the effect of parameters on the case of depth. The efficiency and simplicity of factorial experiments make it the most commonly used model for choosing factor levels and the simultaneous mode of factor variation in order to study the effect of each factor on the process [22]. In the case of laser hardening processes, it is important to define the experimental margins by allowing for minimal case depth transformation and excessive transformation. Maximal and minimal tuning is necessary before parameter levels can be established. The experimental strategy consists of investigating independent and controllable operational factors [23]. The five factors that are used in these experiments are laser power (P), scanning speed (S), rotation speed (R), flank tilt angle (A), and tooth depth (D). Table 3 shows the factors and their variation ranges in our experiments. Four factors with three variation levels were considered for laser power, scanning speed, rotation speed, and tooth depth. The remaining factor, flank tilt angle (A), had only two variation levels due to the non-availability of manufacturing tools. All the factors used in these experiments are continuous and interrelated. The average variation range of laser hardening parameters, which allows us to achieve an acceptable case depth in a laser heat treatment process that is determined as 2200 W, 4 mm/s, and 2000 rpm.

**Table 3.** Factors used in the experimental tests and range of variables.

| Factors | Abbreviation | Variation Ranges | Unit |
| --- | --- | --- | --- |
| Laser power | P | 1900, 2200, and 2500 | W |
| Scanning speed | S | 2, 4, and 6 | mm/s |
| Rotation speed | R | 1500, 2000, and 2500 | rpm |
| The flank tilt angle of spline tooth | A | 15 and 20 | Degree [°] |
| Tooth depth of spline shaft | D | 2.5, 3, and 3.5 | mm |

Each test consists of linear displacement of the spot in the spline's medium plane. Using a micro-hardness measuring machine, the hardness of heat-treated splines is measured by examining their cross-section to determine the hardness profile (case depth). These results are then exploited using the contributions and average effects of each studied factor on the final response. To this end, the analysis of variance (ANOVA), a statistical model which analyzes the differences among group means in a sample, is used to determine the effect of each factor on the case hardening of splines. The percentage contribution of a single factor reflects the total variation observed in the experiments attributed to this factor and the interactions between factors [24]. These analyses are then followed by performing a linear regression to obtain a mathematical model of case hardening as a function of independent variables and their interactions. This mathematical model can predict the case depth of spline shafts at the tip and root of splines' teeth. Lastly, sensitivity analyses of case depths based on the response surfaces [25] as a function of laser hardening process parameters is performed to obtain a more clear understanding of the estimated model responses. The sensitivity study is established to explore the effects of the laser hardening process parameters on the hardness profile. In fact, the response surfaces present contour plots of case depths at the tip and root of a splines' tooth with respect to the variation of the process parameters.

In Section 3, the results of case depth measurements are presented and supported with ANOVA analysis. These analyses allow evaluating the effect of laser treatment parameters on case depth. The predictive model of case depth at tip and root of spline shafts is then developed based on a linear regression method.

## 3. Results and Discussions

### 3.1. Hardness Measurement

As explained in Section 2, the hardness of heat-treated splines is measured at their cross-section to determine the hardening profile (case depth). The grid of tests in Table 4 presents the measurement of case depth at tip ($H_T$) and root ($H_R$) of splines. The experimental tests consist of varying factors of the laser hardening process, which can affect $H_T$ and $H_R$. Based on the literature presented in Section 1, we selected five parameters in which the modeling factors are: laser power (P), laser beam scanning speed (S), rotation speed of splines during laser heating process (R), flank tilt angle of splines (A), and tooth depth of splines (D). The results obtained show that the case depths are minimal in Test 2 (1900 W, 4 mm/s, 2000 rpm, and 15°) and maximal in Test 9 (2500 W, 2 mm/s, 2500 rpm, and 20°). The case depth is 0.05 mm at the tip and 0.05 mm at the spline's tooth in Test 2, while the case depth is 0.95 mm at the tip and 0.45 mm at the root of the spline's tooth in Test 9.

**Table 4.** Experimental test results with respect to factors variation in each test.

| Test | Factors | | | | | Responses | |
|---|---|---|---|---|---|---|---|
| | P [W] | S [mm/s] | R [rpm] | A [°] | D [mm] | $H_T$ [mm] | $H_R$ [mm] |
| 1 | 1900 | 2 | 1500 | 15 | 2.5 | 0.20 | 0.15 |
| 2 | 1900 | 4 | 2000 | 15 | 3.0 | 0.05 | 0.05 |
| 3 | 2200 | 6 | 2500 | 15 | 3.0 | 0.05 | 0.10 |
| 4 | 2500 | 4 | 1500 | 15 | 3.5 | 0.20 | 0.20 |
| 5 | 2200 | 2 | 2000 | 15 | 3.5 | 0.60 | 0.30 |
| 6 | 2500 | 4 | 1500 | 15 | 3.0 | 0.25 | 0.20 |
| 7 | 1900 | 6 | 1500 | 15 | 3.0 | 0.05 | 0.05 |
| 8 | 1900 | 4 | 2500 | 15 | 2.5 | 0.05 | 0.05 |
| 9 | 2500 | 2 | 2500 | 20 | 3.0 | 0.95 | 0.45 |
| 10 | 2200 | 4 | 1500 | 20 | 2.5 | 0.15 | 0.15 |
| 11 | 1900 | 6 | 1500 | 20 | 3.5 | 0.05 | 0.10 |
| 12 | 1900 | 4 | 2000 | 20 | 3.0 | 0.05 | 0.10 |
| 13 | 1900 | 2 | 1500 | 20 | 3.0 | 0.20 | 0.15 |
| 14 | 1900 | 4 | 2500 | 20 | 3.5 | 0.05 | 0.10 |
| 15 | 2500 | 6 | 2000 | 20 | 2.5 | 0.10 | 0.15 |
| 16 | 2200 | 4 | 1500 | 20 | 3.0 | 0.15 | 0.15 |

The result of hardness curves at the tip and root of the spline's tooth for test 5 is shown in Figure 3a. This hardness carves can be characterized by three regions. The first region demonstrates high surface layer hardness compared to the core hardness of the spline. This is due to a complete austenitization, as a result of high temperature (above $Ac_3$) in the surface layer, and then a martensite transformation resulted by rapid cooling. The second region presents a hardness drop to approximately the core hardness. The temperature in this region is between $Ac_1$ and $Ac_3$, which results in a mixture of hard and over-tempered martensite. The third region keeps the initial hardness of the part since it is not affected by the thermal flow. Therefore, the case depth is defined by the first region, which represents the hardness profile of the surface layer. It is worth mentioning that the case depth of heat-treated spline at the tip and root of the tooth is presented with red and yellow lines, respectively. The cooling medium after austenitization of surfaces is the air at an ambient temperature ($T_\infty$ = 20 °C). On the one hand, the high heat flux generated by laser beam energy, applied in a localized region, transforms the surface layer to austenite. On the other hand, AISI 4340 steel is a very good heat conductor. Thus, cooling it in the air results in self-quenching of the heated surface. The cooling rate of heated AISI 4340 in the air is rapid enough to eliminate the need for external quenching for martensite transformation.

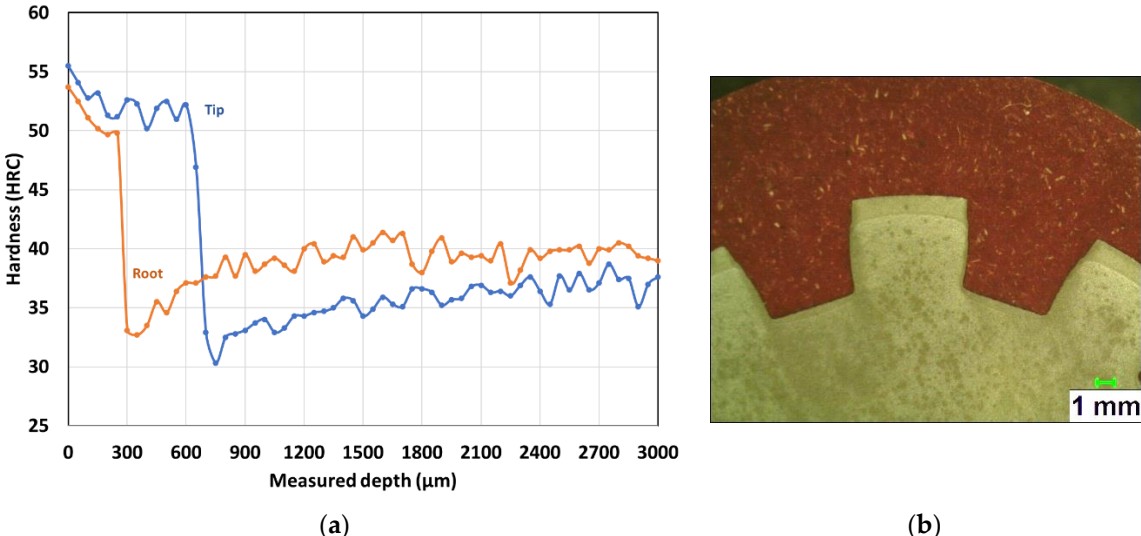

**Figure 3.** Hardness results for laser heat treatment for test 5: (**a**) micro-hardness measurement [HRC] in different depths [μm] at tip and root of spline's tooth and (**b**) visualized hardness profile of a cross-section of the spline shaft.

It appears that the case depth is more present at the tip than at the root of splines. This difference is important in the case of high laser power and low scanning speed. It is interesting to note that the transition region between the hardened layer and the layer below it had minimum hardness with respect to the initial hardness at the surface and the depth. This transition region did not receive enough heat flux to be transformed into austenite. Yet, it is affected by the heat of a hardened surface to which its hardness drops to the original value.

The metallographic analysis of the hardened region reveals a martensitic microstructure, which is illustrated as a shadow layer, in Figure 3b, on the surface of the spline for test 5. The hardened layer follows the shape of the spline's tooth with a circular form, recalling the initial shape of the rotating shaft. The results confirm that it is technically possible to control the case depth during the laser heat treatment process, according to the specific requirements of the manufacturing process for mechanical parts, such as pump motor shaft splines. The case depth at the tooth flank is not affected by the variations of the heat treatment process parameters.

## 3.2. Statistical Analysis

In order to assess the significance of each factor in laser hardening of splines based on our experimental tests, an ANOVA analysis is performed. These analyses are ensured using MINITAB18 statistical analysis software in which their general systematic methodology is detailed in Tables 5 and 6. For each parameter, the value of variance ratio F is compared to values of the standard F tables for given statistical significance levels. It is concluded that the case depth is influenced to a large degree by laser power (P), scanning speed (S), and the interactions between these factors, which were significant for the response value of the model at a 98% confidence. The coefficient of determination ($R^2$), which can be interpreted as the rate of factors contributed in the estimated statistical model, is approximately 99% for case depth at the tip of splines ($H_T$) and 98% for case depth at the root of splines ($H_R$). It can also be observed that the hardness value increases with increasing laser power and decreasing the scanning speed. This parameter combination results in higher absorption of energy in the surface layer of splines, which increases the depth of heat-treated material and, eventually, the case depth. It is also observed that the variation of the flank tilt angle and tooth depth of the spline does not noticeably affect the case depth of splines.

**Table 5.** ANOVA table for case depth at the tip of the splines ($H_T$).

| Factors | DoF | Sum of Squares | Mean Square | F-Value | P-Value |
|---------|-----|----------------|-------------|---------|---------|
| P | 1 | 0.229105 | 0.251229 | 290.93 | 0.001 |
| S | 1 | 0.361250 | 0.015161 | 17.56 | 0.004 |
| R | 1 | 0.036946 | 0.009494 | 10.99 | 0.013 |
| A | 1 | 0.003906 | 0.003906 | 4.52 | 0.071 |
| D | 1 | 0.020000 | 0.002943 | 3.41 | 0.107 |
| $S^2$ | 1 | 0.097656 | 0.097656 | 113.09 | 0.001 |
| P × S | 1 | 0.137336 | 0.138967 | 160.93 | 0.001 |
| D × S | 1 | 0.005100 | 0.005100 | 5.91 | 0.045 |
| Error | 7 | 0.006045 | 0.006045 | – | – |
| Total | 8 | 0.897344 | – | – | – |

**Table 6.** ANOVA table for case depth at the root of splines ($H_R$).

| Factors | DoF | Sum of Squares | Mean Square | F-Value | P-Value |
|---------|-----|----------------|-------------|---------|---------|
| P | 1 | 0.056378 | 0.045650 | 209.35 | 0.001 |
| S | 1 | 0.045000 | 0.001380 | 6.33 | 0.033 |
| R | 1 | 0.003196 | 0.001418 | 6.50 | 0.031 |
| A | 1 | 0.001406 | 0.001406 | 6.45 | 0.032 |
| $S^2$ | 1 | 0.012656 | 0.012656 | 58.04 | 0.001 |
| P × S | 1 | 0.020495 | 0.020495 | 93.99 | 0.001 |
| Error | 9 | 0.001962 | 0.020495 | – | – |
| Total | 15 | 0.141094 | 0.001962 | – | – |

*3.3. Contribution and Effect of Process Parameters*

The statistical approach allows determining the main effects of each factor separately as well as the different interactions between these factors and the response variable with the same precision. It also facilitates analysis of the experimental data and the development of correlations between studied factors and the case depths. The factors' contribution in the estimated model's performance at the tip and root of splines' tooth is analyzed by the ANOVA variance study. This study allows determining the different effects and the contribution of each factor in the final response. The contribution of each process parameter (P, S, R, A, and D) with a significant interaction between some parameters on the model's final response are presented in Table 7. The results of ANOVA analysis confirm that laser power (P) and scanning speed (S) contribute significantly to the final response of the case depth. It is also observed that rotation speed (R), flank tilt angle of the spline tooth (A), and tooth depth (D) have a small influence on the results in which their contributions are, respectively, of the order of approximately 4%, 1%, and 2%. This is due to their small variation in the process parameters. The overall error of the statistical model is less than 2%.

**Table 7.** Parameters contributions as a percentage [%] in the case depth of splines.

| Parameters | $H_T$ [%] | $H_R$ [%] |
|------------|-----------|-----------|
| P | 25.53 | 39.96 |
| S | 40.26 | 31.89 |
| R | 4.12 | 2.27 |
| A | 0.44 | 1.00 |
| D | 2.23 | 0.00 |
| $S^2$ | 10.88 | 8.97 |
| P × S | 15.30 | 14.53 |
| D × S | 0.57 | 0.00 |
| Error | 0.67 | 1.39 |

The graphical representation of the main effects of all parameters on the case depth (in μm) at tip ($H_T$) and root ($H_R$) of splines are shown, respectively, in Figures 4 and 5. The results illustrate the qualitative contribution of different factors to the variation of $H_T$ and $H_R$ and offer information on the nature of their relationship. The steep variation of case depth with respect to laser power (P) and scanning speed (S) confirms that these factors contribute the most in the final response, which is consistent with the results provided by ANOVA analysis. These results illustrate that the case depths at the tip and root of splines increase as laser power (P), rotation speed (R), flank tilt angle (A), and tooth depth (D) increase. Furthermore, the case depths at the tip and root of splines decrease as the scanning speed (S) decreases.

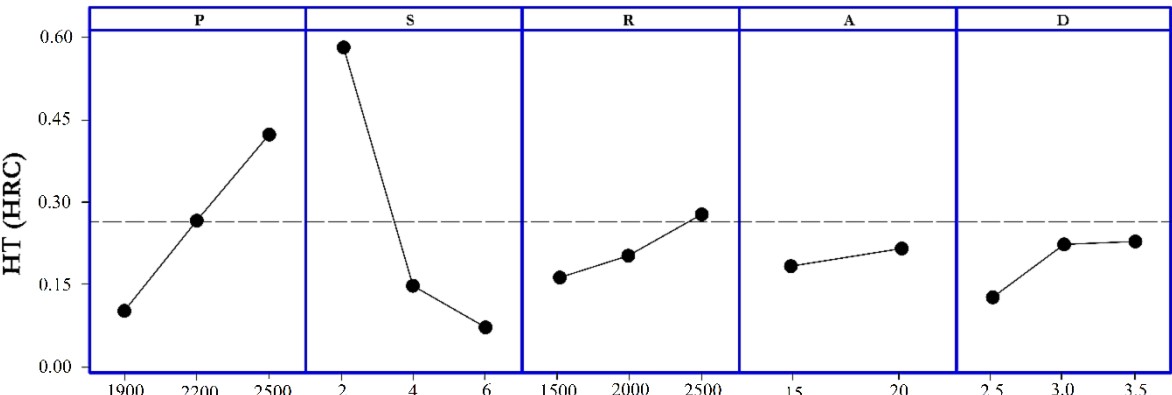

**Figure 4.** Effects of process parameters on the case depth at the tip of the spline's tooth ($H_T$). The unit of parameters are laser power (P) in [W] scanning speed (S) in [mm/s], rotational speed (R) in [rpm], flank tilt angle (A) in degrees, and tooth depth (D) in [mm].

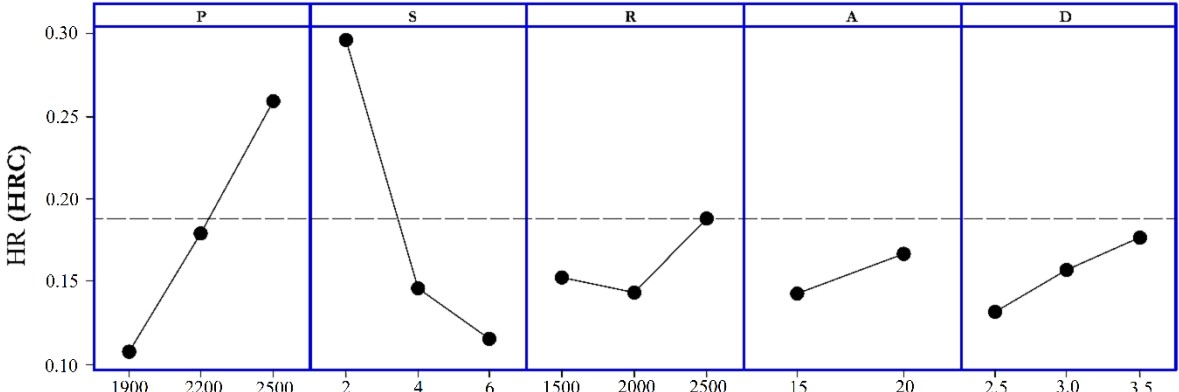

**Figure 5.** Effects of process parameters on the case depth at the root of spline's tooth ($H_R$), the unit of parameters are laser power (P) in [W] scanning speed (S) in [mm/s], rotational speed (R) in [rpm], flank tilt angle (A) in degrees, and tooth depth (D) in [mm].

To estimate the case depths at tip ($H_T$) and root ($H_R$) as a function of laser power (P), scanning speed (S), rotation speed (R), flank tilt angle (A), and tooth depth (D), a linear regression technique is used to determine the adequate coefficients in a polynomial function. After eliminating non-significant terms, the empirical estimated relationships of case depth at tip ($H_T$) and root ($H_R$) of a tooth can be expressed according to the following equations.

$$H_T = -3.122 + 0.001717P + 0.3953S + 0.000065R + 0.00625A + 0.1373D + 0.03906S^2 - 0.000319(P \times S) - 0.0452(S \times D) \quad (1)$$

$$H_R = -0.981 + 0.000649P + 0.0680S + 0.000023R + 0.00375A + 0.01406S^2 - 0.000103(P \times S) \quad (2)$$

The measured and predicted curves of the statistical model responses at the tip and root of the splines' tooth are presented in Figures 6 and 7, respectively. In these figures, the measured data represents the experimental tests presented in Table 2. It can be observed that the estimated case depth at the tip and root of the splines' tooth within the range of parameters in this study is estimated with a good accuracy wherein the estimation error is less than 5%. High laser power and low scanning speed lead to a higher case depth in Test 9, which explains the response model's peak in the curve. The predicted curves are aligned with the measured curves, which explains the correspondence between the estimated and measured case depth values.

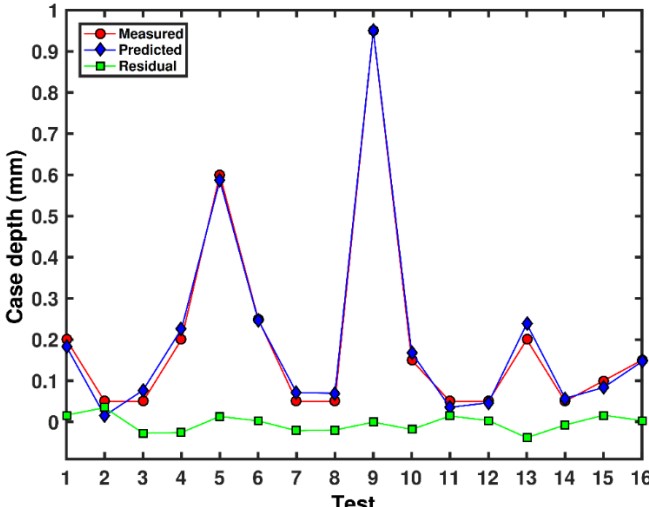

**Figure 6.** Measured values of case depths [μm] at the tip of the splines' tooth versus the predicted values.

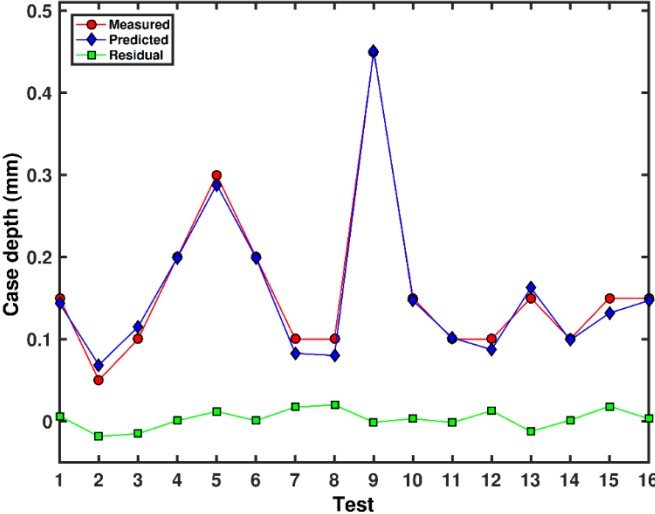

**Figure 7.** Measured values of case depths [μm] at the root of the splines' tooth versus the predicted values.

*3.4. Response Surfaces and Sensitivity Analysis*

As mentioned in Sections 3.2 and 3.3, among the process parameters, only laser power (P) and scanning speed (S) have a dominant effect on the hardening process and the case depth of splines. Therefore, the sensitivity analyses of these two parameters can illustrate the general trend of splines' case depth with respect to the variation of laser power and scanning speed. To this end, the estimated response surfaces, as contour plots, of case depth at the tip and root of splines' tooth versus the evolution of laser power and scanning speed are presented in Figures 8 and 9. To reduce the number

of statistical response surfaces, only the mean value of rotation speed (R = 1875 rpm) and tooth depth of spline (D = 3 mm) is taken into consideration as a constant parameter in the contour plots. The effect of tooth depth variation on the case depth at the root of a spline's tooth is negligible. Therefore, this parameter is not taken into consideration in the contour plot of case depth at the root ($H_T$). In the case of the flank tilt angle of splines (A), which is the last parameter of the process, both angles (15° and 20°) are separately taken into consideration in the contour plots of case depth at the tip and root of the splines. The contour plots of case depth at the tip and root with a flank tilt of 15° are presented in Figure 8, and those with a flank tilt of 20° are presented in Figure 9. These figures show the effects of the laser process parameters (P, S, and R) and geometrical factors (A and D) on the case depth of splines. In these figures, the unity of the independent variables and parameters are presented as follows: the case depth in [mm], laser power in [W], scanning speed in [mm/s], and rotational speed in [rpm].

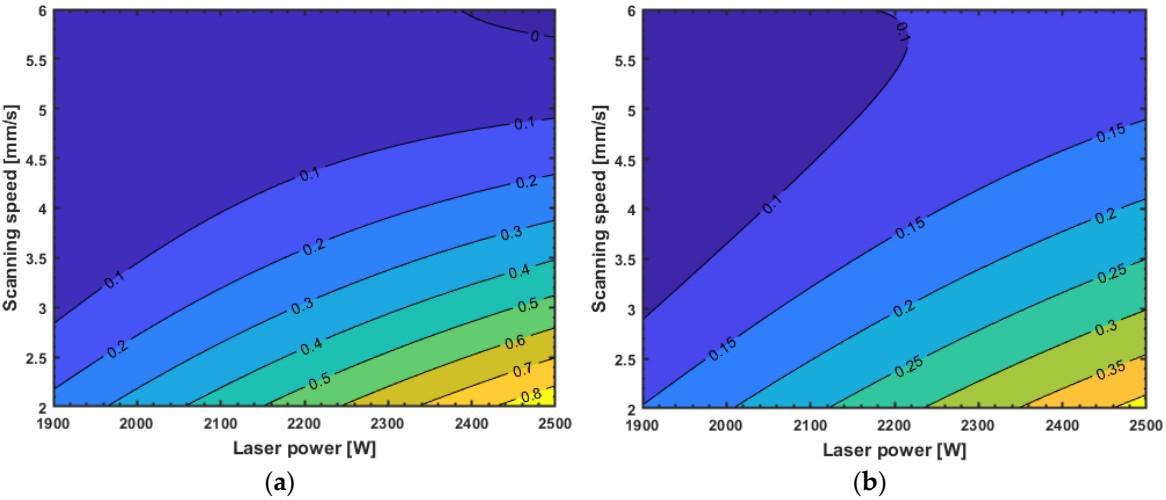

**Figure 8.** Contour plot of case depth [mm] versus power, scanning speed and rotation speed (1875 rpm) for the flank tilt angle of 15°: (**a**) at the tip of splines' tooth ($H_T$), (**b**) at the root of splines' tooth ($H_R$).

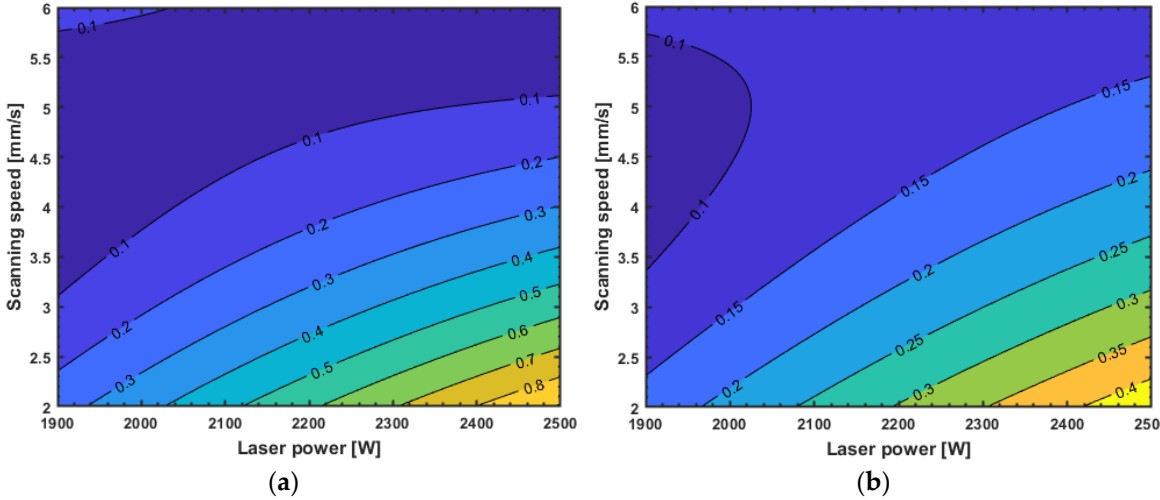

**Figure 9.** Contour plot of case depth [mm] versus power, scanning speed, and rotation speed (1875 rpm) for the flank tilt angle of 20°: (**a**) at the tip of splines' tooth ($H_T$), (**b**) at the root of splines' tooth ($H_R$).

It can be observed in Figures 8 and 9 that the case depth at the tip and root of splines reacts directly with laser power (P) and inversely with scanning speed (S). In other words, increasing the laser power and decreasing the scanning speed results in increasing the case depth regardless of the tilt angle of the splines' tooth (15° or 20°). The higher case depth is generally obtained at the tip and root of the splines'

tooth corresponding to the flank tilt angle of 20°, which is a very representative fact since the surface of the splines' tooth can be exposed better to a laser beam where the flank tilt angle is 20° when compared to 15°.

## 4. Conclusions

The effect of different parameters in a laser hardening process on the case depth of AISI 4340 steel splined shafts is studied in this article. Laser hardening is a heat treatment process that allows microstructural transformation on the surface layer of steel. This microstructural transformation generates martensite on the surface layer of AISI 4340 splines, which increases their surface hardness. This process, which is also known as case hardening, results in a hard and wear-resistant surface while maintaining a soft and ductile core. The study is based on developing experimental tests by varying the hardening process parameters as well as the geometry of splines. These tests are then analyzed using statistical approaches to define the effectiveness of each parameter on the case depth at the tip and root of splines' tooth.

The influence of laser power, scanning speed, tooth depth, and flank tilt angle on the laser hardening process of spline shafts with different dimensions are studied and analyzed in this study. Performing experimental tests wherein test parameters are systematically varied and then using ANOVA analysis demonstrate that the case depth is mainly affected by the factors of laser power and scanning speed. These factors contribute to approximately 70% of case depth variability at the tip and root of splines' tooth. In addition, a statistically estimated model of case depth at the tip and root of splines is proposed based on correlations between case depths and laser power, scanning speed, rotation speed, flank tilt angle, and tooth depth. These correlations estimate the case depth of splines with an accuracy in which the mean square error is less than 5%. The obtained results can be used to plan the modeling process and be served as a development guideline based on case depth predictions for AISI 4340 steel splines with studied geometries.

Even if the validation results of these predicted models are promising, they can be improved by performing additional tests outside the variation range of parameters presented in this study. This ensures the validation of our predictive models on a larger range of parameter variation. Applying an artificial neural network based on the results of these tests can also improve the estimation of predictive models. Lastly, a statistical approach based on validation and verification methods can be applied to evaluate quantitatively the precision of predictive models.

**Author Contributions:** Conceptualization, N.B.; methodology, N.B., S.S.K. and R.F.; software, R.F, P.T. and V.B.F.K.; validation, S.S.K. and N.B.; formal analysis, N.B., S.S.K. and R.F.; investigation, S.S.K.; resources, R.F., P.T. and V.B.F.K.; data curation, R.F., P.T. and V.B.F.K.; writing—original draft preparation, R.F. and S.S.K.; writing—review and editing, S.S.K. and N.B.; visualization, R.F., S.S.K. and N.B.; supervision, N.B.; project administration, N.B.; funding acquisition, N.B. All authors have read and agreed to the published version of the manuscript.

**Funding:** This research received no external funding.

**Conflicts of Interest:** The authors declare no conflict of interest.

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
