# Peer review of "Effects of Laser Hardening Process Parameters on Hardness Profile of 4340 Steel Spline—An Experimental Approach"

_coatings, doi:10.3390/coatings10040342_

Round 1
Reviewer 1 Report
In the paper the hardening of splined shaft by means of laser treatment was studied. The case depth was estimated based on hardness profile measurements on the tip and the root of the splined shaft teeth. The sensitivity of the case depth on evolution of hardening process parameters was analysed. The model was proposed that that enables to estimate the case depth as a function of laser power, scanning speed, rotation speed, flank tilt angle and tooth depth. The approach presented by the Authors is interesting, however some points should be clarified.
- It is not clear why the function H/R x (P/S)1/3 (Equation 6) is dimensionless ?
- The quantity presented on axis X in Figs. 8, 9 (P/S R3) 1/3 does not correspond to a dimensionless function
- The proposed model, Eqs. 1, 2, were verified only for the experimental points, that were used to calibrate this model. It would be good to show some results of model and experiment for other points
- One can conclude from Fig. 3, that a resolution of the case depth measurement is approximately 0,05mm, probably due to required minimal distance between hardness impressions. On the other hand in the Table 2 we have many points corresponding to 0.05mm case depth. An exact specification of case depth for these points is therefore questionable, since a measured variable should be greater than the measurement resolution.
Reviewer 2 Report
The manuscript is interesting, but it will be useful to provide some explanation for the reason for the effect of the different parameters or at least for some of them. The manuscript needs some improvement as suggested below:
- Whilst it is common to have reference numbers at the end of a sentence, in those cases where the name of the author is mentioned, the reference number needs to follow the name, i.e., in line 42, the sentence should start as follows: Lambiase et al [12] ...... Also in line 39, Sun et al should be followed by the reference number.
- The headings of some of the sections must change. For example, the second section should be called Methodology and describe the experimental methods, etc. and the third section should present results, etc. In the manuscript the results are in the second section and this must be changed.
- The composition of 4340 steel must be presented.
- What heat-treatment was carried out to give 45HRC? The process (annealing and/or tempering), temperature and cooling method must be given.
- To what finish was the material polished? How was it polished?
- In line 113, there is reference to section 0. This seems to be an error.
- Figure 3 must be divided into figure 3(a) and 3(b) and not 3-left and 3-right.
- In line 132, there is mention of a very fine martensitic microstructure, but the magnification is too low for any fine martensite to be seen. Can another figure be shown to present the fine martensite?
- In lines 152-155, it is stated that "The results obtained confirm that the hardness value increases with increasing laser power and rotation speed while the hardness decreases with increasing scanning speed." It will be good to discuss why this is so, i.e., why does the hardness increase with increasing laser power.
- The units of the parameters in figures 4 and 5 are not given.
- In line 198, why does high laser power and low scanning speed lead to a higher case depth?
- There is some problem in line 231 regarding a reference which seems missing.
Reviewer 3 Report
- Original Submission
1.1. Recommendation
Minor Revision
- Comments to Author:
Ms. Ref. No.: Coatings-706975-peer review
Title: Effects of laser hardening process parameters on hardness profile of 4340 steel spline – an experimental approach
Noureddine Barka, Sasan Sattarpanah Karganroudi, Rachid Fakir, Patrick Thibeault, Vincent
Blériot Feujofack Kemda
Overview and general recommendation:
This paper investigates the effect of different parameters in a laser hardening process on the case depth of 4340 steel shaft splines based on experimental tests and statistical analysis. The influence of laser power, scanning speed, tooth depth and flank tilt angle on the laser hardening process of spline shafts with different dimensions are studied and analyzed in this study. Authors concluded that laser power and scanning speed affect directly the case depth. A statistically estimated model is proposed based on correlations between case depth at the tip or at the root of spline and laser power, scanning speed, rotation speed, flank tilt angle and tooth depth.
I found the paper to be written overall well, however, there are a few typo and minor things to be addressed. Also, novelty of this paper is still not clear. Therefore, I recommend that a minor revision is warranted. Below are my comments:
- Page 1: Line 33 – Please mention what parameters are considered in general like power, scan speed etc.
- Figures are not sharp.
- Font size in the figures is inconsistent
- Figure 2: df means degrees of freedom? It was abbreviated as DoF in page 2 but used as df. Also, which robot was that. Any model number?
- Conclusions: Please check the grammar. Few typos and phrases are redundant
- Please check format of references. It is inconsistent
I hope these comments are useful to refine the paper. Please feel free to ask any questions if you have.
Round 2
Reviewer 1 Report
The response of Authors and corrections made in the paper are satisfactory